# Psychological price perception may exert a weaker effect on purchasing decisions than previously suggested: Results from a large online experiment fail to reproduce either a left-digit or perceptual-fluency effect

**Achiel Fenneman**[1,2,3]*, **Jörn Sickmann**[2], **Sascha Füllbrunn**[1], **Carina Goldbach**[2], **Thomas Pitz**[2]

**1** Institute for Management Research, Radboud University, Nijmegen, Netherlands, **2** Faculty Society and Economics, Rhein-Waal University of Applied Sciences, Kleve, Germany, **3** Donders Institute for Neuroimaging, Radboud University, Nijmegen, Netherlands

* a.fenneman@gmail.com

**Data Availability Statement:** The preregistration, custom software used during the experiment, as

## Abstract

Retail store prices are frequently set to either a just-below (e.g., $1.99) or rounded to (e.g., $2.00) integer levels. Previous studies proposed two price-perception effects that may underly such psychological pricing strategies. First, the left-digit effect (LDE) assumes consumers read prices left-to-right. Cognitive limitations let consumers overweight the impact of digits on the left side of the price while underweighting digits on the right side of the price. This effect appears to conflict with the contradictory perceptual fluency effect (PFE), which proposes that a rounded price is more perceptual fluent and, thus, more attractive to consumers. To address these paradoxical effects, we conducted an online experiment with 266 participants making a total of 4788 purchasing decisions where we systematically varied the purchasing prices of otherwise identical lottery tickets across two price levels. Against expectations, we found no support for either of the two price-perception effects. We propose three possible explanations of these null results.

## Introduction

A trip to any retail venue through the West reveals an odd peculiarity: a substantial number of prices are either a rounded integer price (e.g., $2.00) or just below an integer amount (e.g., $1.99). Of these numbers, the just-below prices are substantially more common. Studies in the late '90s reported that up to 65% of all UK retail prices were such 99-ending prices [1–3]. A more recent report conducted for the Deutsche Bundesbank [4] observed that a total of 62.9% of a large sample of German retail prices were 99-ending prices. Moreover, out of the 20 most frequently observed retail prices, no less than 18 had 9-ending prices. While these just-below prices are the most frequently occurring prices, rounded 00-ending and 5-ending prices account for a large chunk of the remaining prices [2, 3]. Furthermore, the relative frequency of

well as the (anonymized) data and the analysis script are freely available online via https://osf.io/yxbu8

**Funding:** The author(s) received no specific funding for this work.

**Competing interests:** The authors have declared that no competing interests exist.

such rounded prices has increased in recent years [5]. The overabundance of such rounded and just-below prices is far larger than can be expected to occur by random chance alone. Clearly, something is 'special' about these pricing strategies, but what is it? Previous research suggests two psychological effects of interest: the left-digit effect and the perceptual fluency effect.

On the one hand, as Western numerals are read (and assumed processed) left-to-right, the consumer is first exposed to the leftmost digits of a printed price. Suppose consumers have limitations in their cognitive capability. In that case, this left-first processing might be vulnerable to an anchoring heuristic [6], resulting in an over-estimated impact of the leftmost digits of an observed price. Such overweighting results in a lower, subjective valuation of the overall product. This cognitive peculiarity became the 'left-digit effect' (**LDE**; [7]). Retailers can exploit this processing order by slightly reducing the price of a product such that its first digit slightly crosses an integer threshold (e.g., $1.99 versus $2.00). A wide range of empirical findings supports the LDE. A range of natural experiments and field studies suggest that companies observe an increase in revenue after adopting a just-below pricing strategy [8–11]. Furthermore, in laboratory experiments, price discounts are erroneously remembered as higher if the sale price is set to a just-below number [12], and experimental participants have a larger probability of overspending on a fixed budget when faced with just-below prices than with rounded prices [13].

On the other hand, rounded prices may impart a positive affective judgment as well. Products with a high processing fluency are judged to be more attractive than products with a low processing fluency [14, 15]. As rounded prices are easier to process, they "*feel right*" [16]. This research supports what we dub to be a *perceptual fluency effect* (**PFE**), in which the perceived perceptual fluency of rounded price products (e.g. $2.00) results in a higher attractiveness than identical products with a non-rounded price (e.g., $1.99 or $2.01). A range of previous research supports such a PFE. When consumers self-select their prices, they gravitate towards fluent prices [17]. In the context of pay-what-you-want schemes, consumers select rounded amounts and typically set their gratuity such as to summate to a round total. Finally, the increased usage of rounded prices in retail has been suggested to result from higher confidence experienced by consumers [5], causing them to develop a more favourable overall opinion of the product.

Paradoxically, while both the LDE and PFE are well-supported by previous research, they conflict in their predictions. The LDE suggests consumers find the just-below prices more attractive, while the PFE suggests consumers find the rounded prices more attractive. At first glance, these theories appear mutually exclusive. However, the strong empirical support for both theories suggests that they both provide at least a partial explanation for the efficacy of psychological pricing strategies.

How can this paradox be resolved? First, it is possible that the LDE and PFE both exist but partially offset each other. Alternatively, there is support for a differential effect of both the LDE and PFE at different price levels, with previous evidence suggesting that the LDE dissipates over larger values of the purchasing price [12, 18–20]. Under this interpretation of the paradox, the LDE primarily skews price perception at low prices when the effect of an overweighted first digit is largest. For higher price levels, the relative impact of this skew dissipates. In contrast, perceptual fluency may exert a more constant effect, independent of the overall price level. As a result, the LDE may be dominant at low prices, while the PFE is dominant at higher price levels.

While such a multiple-effects interpretation could account for both effects, a second possible interpretation pertains to the methodological setup in the studies outlined above. In the typical price perception study, participants observe an advertisement (or series of ads) and

state their willingness to purchase the advertised product at a given price. As these studies utilize such an un-incentivized vignette methodology, they are potentially limited in their generalizability to real-world decisions with real-world stakes. Furthermore, prior studies have relied on a moderate sample size, which is likely to result in low statistical power–in turn leading to an overstated effect size of any published effect sizes (for more details, see [21]). In related fields of research, a combination of small sample sizes and a publication bias toward statistically significant results are contributing factors to the recent replication crisis, most notably in the fields of psychology [21, 22] and cognitive neuroscience (e.g. [23]).

We address both the inconsistency in the literature as well as the methodological concerns by conducting a (preregistered) controlled online experiment with a large sample size (N = 266 participants, completing a total of 4788 purchasing decisions). The participants decided whether or not to purchase lottery tickets with the ticket price saliently displayed on the screen. This setup provides a best-case scenario for both the LDE and PFE: participants repeatedly made purchasing decisions with a prominent price, few non-price contextual factors to distract from the price and a relatively low monetary impact.

Against expectations, our results fail to replicate either the PFE or LDE. When keeping the overall attractiveness of tickets fixed, neither tickets with a just-below price (e.g., 1.99), nor tickets with a rounded price (2.00) were purchased at a (statistically significant) higher rate than tickets in a control treatment that were neither rounded nor just-below an integer amount (e.g., 2.01). We provide three possible interpretations of these null results. Both price-perception effects may have a weaker impact than previously described. The effects may be more pronounced when two or more items are competitively compared. Finally, consumers may have 'caught' on to these psychological pricing strategies and developed compensatory purchasing heuristics–that is, while the mechanisms that underlie both pricing strategies still exsits, consumers may have adapted a set of decision rules aimed at mitiating these mechanisms (i.e, "round each price to its nearest integer").

## Methods

### Materials

Each participant made 21 separate purchasing decisions presented in random order. During each decision, the participant could decide whether or not to purchase a lottery ticket. Each lottery ticket had a stated price and two possible outcomes, which were equally likely to occur (an example is given in Fig 1). All prices and outcomes were denominated in Tokens, the experimental currency unit used throughout the experiment (1 Token = 0.10 USD).

We implemented a 2x3x3 factorial within-subjects design with the dimensions 'price level' (high = 9 Token vs low = 2 Token), 'price-ending' (just-below, rounded or control), and 'attractiveness' (low, mild or high), resulting in 18 lottery tickets to represent all 18 combinations.

The price-ending treatments provided a discriminatory test for the LDE and PFE. For each lottery, we determined the lottery's price $p = price\ level + \delta$. For the 'rounded' treatment, we set $\delta = 0$, corresponding to a price of either 2.00 or 9.00 Tokens. For the 'just-below' treatment, we set $\delta = -0.01$, corresponding to a price of either 1.99 or 8.99. Finally, for the 'control' treatment $\delta$ equalled a randomly drawn value between 0.01 and 0.14. We omitted values of 0.05 and 0.09, as these may be considered partially fluent or just-below. Note that ticket prices in the control treatment always had a larger first digit than tickets in the just-below treatment.

We next ensured that each lottery had an identical payoff profile (and expected value) by adjusting the high and low outcomes. The low outcome ('tails') was $0.04 + 0.02\ x\ price\ level + \delta$, while the high outcome ('heads') was $0.04 + \alpha\ x\ price\ level + \delta$. The multiplicator $\alpha$ depended

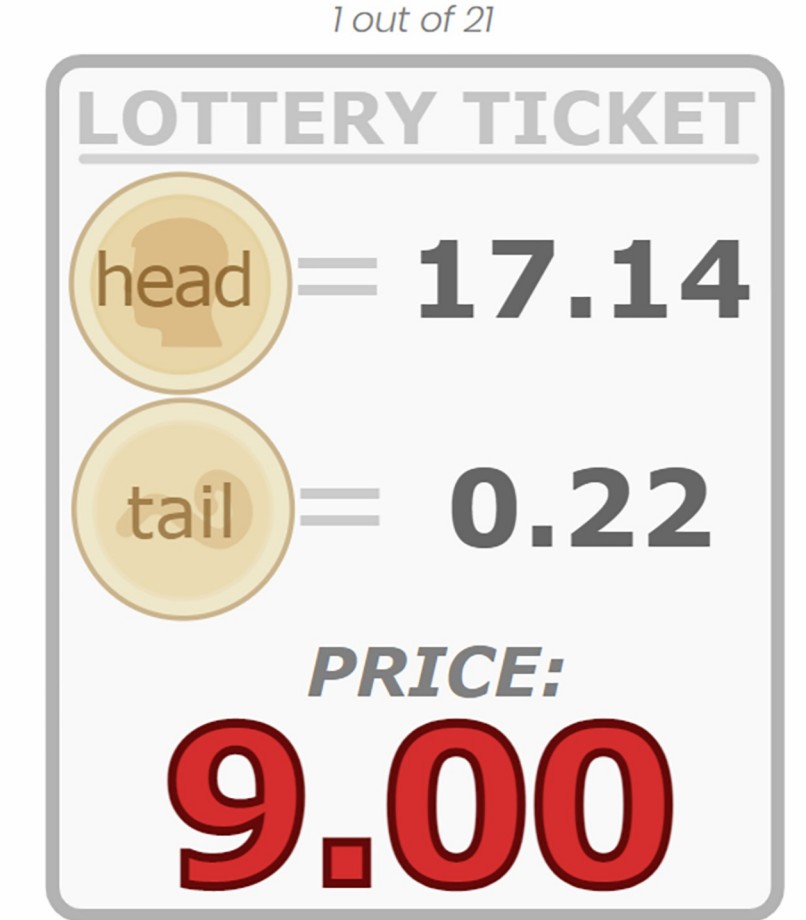

**Fig 1. Example lottery ticket.** Each lottery ticket had a price tag; 9.00 Token in this example. Purchasing a lottery ticket had the following two results: either the lottery ended head, then the payment was 10.00–9.00 + 17.14 = 18.14, or the lottery ended tail, and the payment was 10.00–9.00 + 0.22 = 1.22. Participants could indicate their decision by clicking on one of the two buttons below the text. All lottery tickets were presented to participants sequentially in randomized order. Only one decision was paid out.

on the attractiveness treatment. We consider $\alpha = 1.9$, $\alpha = 2.1$ and $\alpha = 2.3$ for the attractiveness treatments 'low', 'mildly', and 'high', respectively. We added the 0.04 Tokens to prevent outcomes from being either perceptually fluent or just-below. Table 1 provides an overview of the resulting lottery tickets.

We additionally added three lottery tickets to function as attention checks. Each of these lottery tickets was given a randomly selected price level and price-ending treatment. However, all of these lottery tickets had a high outcome that was 0.9 times the ticket price. As a result,

**Table 1. Overview of the 18 lotteries-of-interest.**

| Price level | Attractiveness | Rounded (δ = 0) | | | just-below (δ = -0.1) | | | Control (δ = random) | | |
|---|---|---|---|---|---|---|---|---|---|---|
| | | price | tail | head | price | tail | head | price | tail | head |
| Low (2.00) | Low (α = 1.9) | 2.00 | 0.08 | 3.84 | 1.99 | 0.07 | 3.83 | 2.06 | 0.14 | 3.90 |
| | Mild (α = 2.1) | 2.00 | 0.08 | 4.24 | 1.99 | 0.07 | 4.23 | 2.09 | 0.17 | 4.33 |
| | High (α = 2.3) | 2.00 | 0.08 | 4.64 | 1.99 | 0.07 | 4.63 | 2.13 | 0.21 | 4.77 |
| High (9.00) | Low (α = 1.9) | 9.00 | 0.22 | 17.14 | 8.99 | 0.21 | 17.13 | 9.11 | 0.33 | 17.25 |
| | Mild (α = 2.1) | 9.00 | 0.22 | 18.94 | 8.99 | 0.21 | 18.93 | 9.02 | 0.24 | 18.96 |
| | High (α = 2.3) | 9.00 | 0.22 | 20.74 | 8.99 | 0.21 | 20.73 | 9.04 | 0.26 | 20.78 |

Participants made purchasing decisions for all 18 lottery tickets outlined above, plus three attention check tickets (not shown). The rows price, tail and head show for each lottery the price, the low outcome and the high outcome, respectively. All lottery tickets were presented in random order. For lotteries in control, the value of δ was randomized for each ticket and differed between participants.

even in a best-case scenario, these lottery tickets would yield a negative return. Under the assumption that no attentive participant would be interested in purchasing these tickets (regardless of risk tolerance levels), we excluded any participants from further analysis if they purchased one or more of these tickets.

## Procedure

The experiment was programmed and conducted on Qualtrics, with the participant being recruited via Amazon Mechanical Turk. We advertised to participants by promising a fixed payment of $0.25 and a variable bonus based on their decisions. Upon entering our landing page, the participants received general instructions about the experiment, containing the name of the academic institution conducting the experiment, estimates of both expected earnings and the duration of the experiment, a short overview of the different stages of the experiment as well as the exchange rate between Tokens and USD. Upon consenting to these terms, the participants continued to the instructions for the lottery task. After participants read these instructions, a comprehension test checked their understanding of the task setup. If participants made errors, they were excluded from the experiment and received a fixed payment. Otherwise, the participant received an endowment of 10 Tokens with which the tickets could be purchased, followed by the 21 lottery decisions.

After completing all lottery decisions, but before the participant received feedback on their payoff, participants were given a partially incentivized questionnaire in the second part of the experiment. To measure numerical ability, we added three incentivized questions from existing numerical ability questionnaires. Question one originiated from [24], questions 2 and 3 from [25]. These three questions were selected from a larger existing set of questions on the basis of a stimulus pilot experiment (N = 488). From this existing set, we maximized the expected between-subject spread in scores by selecting the three questions closest to a 50% answer rate. For each correct answer on this part of the questionnaire the participant received an additional bonus of 5 Tokens. The remaining questions included demographics such as age, gender, highest completed education and income.

At the end of the experiment, one lottery ticket was selected at random to determine the participants' payment (see [26]). If the participant decided to purchase the ticket during this decision, then its price was subtracted from the endowment. The computer then randomly selected one of the two outcomes, and this value was added to the participant's Token account. The endowment provided a sufficient number of Tokens to purchase a lottery ticket, still maintaining an overall positive payoff in case the low outcome was randomly selected. If the

participant decided not to purchase the ticket, then they neither paid the ticket price nor received either of the two outcomes.

## Hypotheses

As the price of lottery tickets in the just-below treatment have a lower first digit, the LDE predicts that participants underestimate their stated price and hence judge the overall attractiveness of these tickets to be higher–resulting in a higher purchasing rate for tickets in this treatment as compared to the other two price-ending treatments. Alternatively, as the prices of lottery tickets in the rounded treatment have a higher perceptual fluency, the PFE predicts that participants hold a higher subjective evaluation of these tickets–resulting in a higher purchasing rate for tickets in this treatment as compared to the other two treatments. If both the LDE and PFE influence participants' purchasing behaviour, then lotteries in both the just-below and rounded treatments are predicted to be more likely to be purchased than tickets in the control treatment. Finally, the PFE may have a differential effect based on the ticket's price level, with the LDE exerting a stronger influence for a lower price level than at higher prices levels, i.e., when the gap between the perceived price and the actual price is larger.

## Preregistration and data availability

We collected participants in batches of 100 until a minimum of 250 valid responses had been collected. A response was valid only if the participant passed the comprehension question and did not buy any attention-check tickets. This minimal sample size was selected in order to ensure sufficient power even for very small effect sizes. We preregistered this sampling procedure, as well as the experimental design and the analysis outlined above.

## Ethical approval

The experimental study received approval from the board of ethics at the Faculty of Society and Economics at the Rhein-Waal University of Applied Sciences. Prior to collection of the data, all participants provided written and informed consent to participation into the study. All participant's information was fully anonymized prior to storage and no deception was employed at any point throughout the experiment.

## Results

A total of 390 unique online participants passed the quiz and completed the experiment. Of these participants, 124 (32%) purchased one or more attention-check. All reported results are robust to the inclusion of these participants. The S1 File includes the main results for the 124 excluded participants separately. No LDE or PFE was observed for this subsample of participants.

The remaining 266 valid participants had a mean age of 39.7 years; 37.2% was female and 61.6% was male (1% selected 'other' or chose not to disclose their gender), 59.8% had a Bachelors degree or higher, 78.9% had an annual income over $30,000 (with 44% earning more than $60,000).

Excluding the attention checks, participants made a total of 4,788 purchasing decisions. We tested our hypotheses via two binomial mixed-effects regressions (Table 2), in both of which the dependent variable was a Boolean variable denoting a ticket purchase (1 = Yes, 0 = No). Both models included participant-level random intercepts as a random effect and the treatment variables as fixed effects. In the first model, we tested the main effect of three price-ending treatments by including two dummies coding for the just-below and rounded price

**Table 2. Mixed-effects regressions.**

| | DV: ticket bought (1 = yes) | |
|---|---|---|
| | **(1)** | **(2)** |
| Just-below Dummy | 0.101 | 0.122 |
| | (0.085) | (0.118) |
| Rounded Dummy | 0.036 | 0.086 |
| | (0.085) | (0.118) |
| Low Price Level Dummy | | -0.094 |
| | | (0.119) |
| Just-below X Low Price Level | | -0.043 |
| | | (0.167) |
| Rounded X Low Price Level | | -0.101 |
| | | (0.168) |
| Constant | **-0.301**\*\* | **-0.254**\* |
| | (0.121) | (0.133) |
| Participants | 266 | 266 |
| Observations | 4788 | 4788 |
| Log Likelihood | -2762.6 | -2764.1 |
| AIC | 5533.1 | 5542.2 |
| BIC | 5559 | 5587.5 |

Regression results for both statistical models. Both models estimated the best-fitting logistic curve to predict the probability that participants purchased a lottery ticket (1 = yes). Each model included participant-level random effects. In the first model we included dummies for both experimental treatments as fixed effects. The second model additionally included fixed-effects dummies for the low price level and the interaction between the price level dummy and the two experimental conditions.

Note

*p<0.1

**p<0.05

***p<0.01.

treatments. The second model additionally included the price level treatment (a single dummy to denote the low price treatment) and the interaction between the price level and the price-ending treatment.

## No aggregate effect of either the LDE or PFE

Fig 2 provides the purchasing rates when aggregating over both price levels and the three attractiveness levels. Participants were slightly more likely to purchase a lottery ticket if its price was either just-below or rounded to an integer amount (47.1% and 45.9%, respectively), as compared to a ticket in the control treatment (45.3%). However, this difference in purchasing rates was minor, and even in our comparatively large sample of decisions, we could not reject the possibility that these results are due to chance (model 1; p > 0.05 for both treatment dummies).

## Results do not support an enhanced LDE at the low price level

Previous research suggested that the LDE may be more pronounced at lower price levels when the relative difference in the first digits is largest between the just-below treatment versus the other two treatments (i.e., a leading digit of 1 versus 2). The two rightmost panels of Fig 2

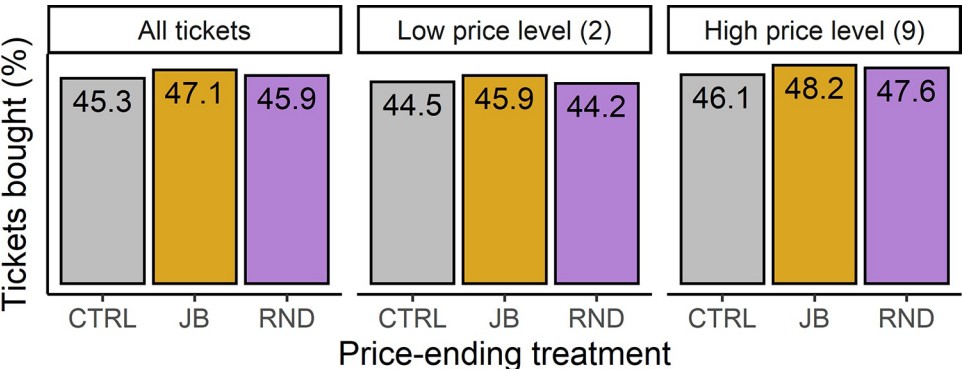

**Fig 2. The main experiment results do not indicate a difference in purchasing rates (y-axis) for the three price-ending treatments. A:** purchasing rates across both price levels. **B:** purchasing rates for the low price level. **C:** purchasing rates for the high price level. CTRL = control prices, JB = just-below prices, RND = rounded prices.

depict the purchasing rates for tickets in the low and high price level treatments, respectively. As before, we observe only a minor increase in purchasing rates for the just-below treatment versus the other two treatments. This difference did not reach statistical significance; we could not reject the possibility that these results were due to chance alone (model 2, p > 0.5 for the interaction between the low price level dummy and the just-below price ending treatment).

## Exploratory analysis of the control variables

In addition to the main treatment variables, we collected a number of exploratory variables. Fig 3 provides a graphical overview of the main results, subdividing participants by their score on the numerical ability questions, education level, age and gender. Due to their exploratory nature, no p-values are provided for these results. Visual inspection of the three plots does not indicate a consistent or substantial effect of either the LDE nor PFE for any of the three exploratory variables. Neither male nor female participants were substantially more likely to purchase tickets in the just-below or rounded price ending treatments (as compared to the control treatment). Furthermore, we did not observe a consistent effect of either age, nor of participant's educational level on their propensity to purchase tickets in either experimental condition (as compared to the control condition).

## Post-hoc power analysis

We did not observe a main effect, so we conducted a posthoc power analysis to determine the least detectable effect size given our experimental methodology. We simulated the hypothetical results for 266 fictive participants who made twelve purchasing decisions: six decisions in a simulated control condition and six in a simulated experimental condition. In line with the observed results, simulated participants had a 45.3% probability of purchasing the lottery ticket in the control condition. We next simulated the results for a range of hypothetical experiments in which participants had a 46% to 53% probability of purchasing tickets for the experimental treatments. Simulations were conducted at 0.1% increments with 2500 simulated experiments per increment to control random noise. In each simulated experiment, we estimated a mixed-effects model to determine the percentage of simulations in which a significant effect was detected. The R-script by which this simulation was conducted is available at https://osf.io/yxbu8.

Fig 4 provides a graphical depiction of the simulated results. Our empirical methodology could reliably (i.e., with a power above 80%) detect a 5% effect size. That is, we had an 80%

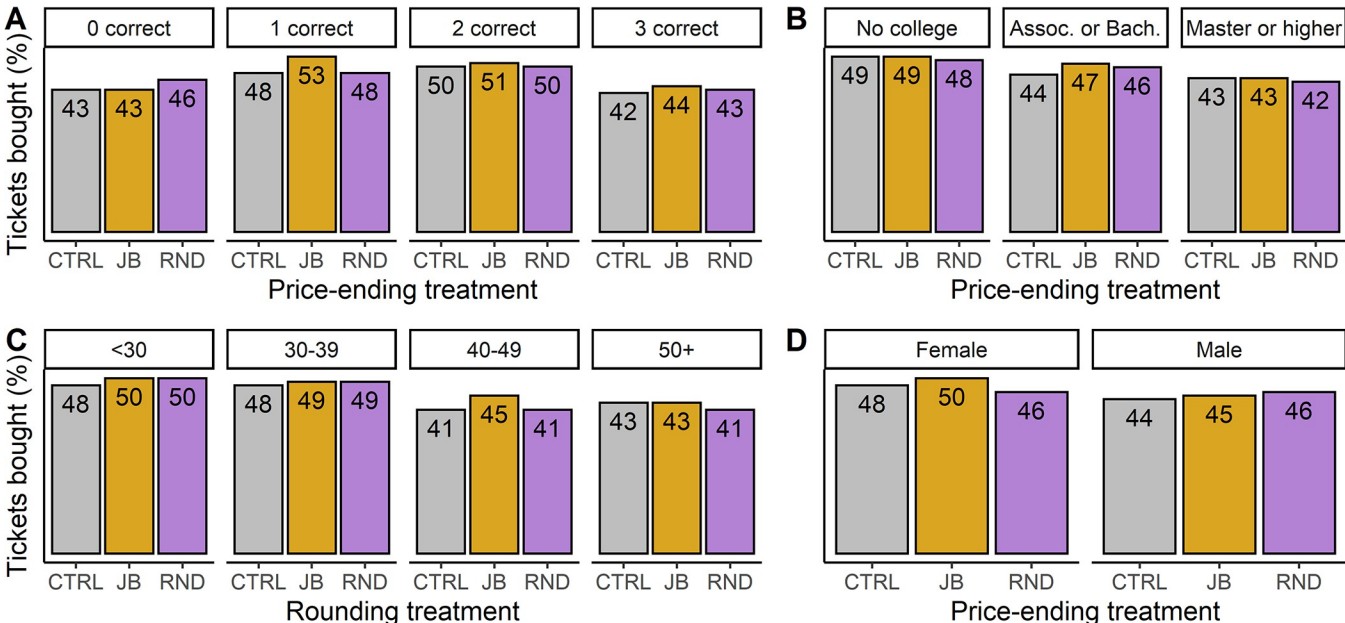

**Fig 3. Purchasing rates across the exploratory variables.** A: purchasing rates divided per numerical ability–i.e., the number of questions the participant correctly answered during the incentivized part of the questionnaire. B: purchasing rates subdivided by highest obtained educational degree. C: purchasing rates subdivided by participant's age. D: purchasing rates by gender (the categories "other" and "do not wish to say" are not depicted due to low sample sizes). CTRL = control prices, JB = just-below prices, RND = rounded prices.

probability of correctly detecting an effect if participants would purchase the experimental ticket in at least 50.25% of decisions. Additionally, we had a 50% probability of detecting an effect if participants were at least 48.75% likely to purchase the experimental tickets.

## Discussion

Our experiment was conducted to resolve the apparent contradiction in the psychological pricing literature, where two proposed effects lead to paradoxical predictions. On the one hand, the *left-digit effect (LDE)* predicts consumers underestimating just-below prices, leading to a higher attractivity of the product. On the other hand, the *perceptual fluency effect (PFE)* predicts that consumers utilize a meta-cognitive strategy in which more easily process-able rounded prices result in a higher attractiveness of a product. In contrast to previous findings, we did not observe significant evidence for either the LDE or the PDE.

These results are inconsistent with previous literature demonstrating either of the two proposed price-perception strategies. At first glance, our findings appear challenging to rhyme with this literature, as our experiment provided a best-case scenario for both effects. Participants made a relatively rapid set of low-stakes decisions, increasing the probability that they relied on a heuristic decision strategy. Furthermore, in all lottery tickets, the price was saliently displayed, and the other features of the lottery tickets were relatively devoid of context. As a result, participants were forced to rely heavily on the ticket's price when evaluating the attractiveness of the lottery ticket. We a-priori created these conditions to maximize the probability of observing both effects, leaving it unclear why we fail to produce either effect. Below we propose a number of possible explanations for this disparity between our findings and the previous literature. However, none of these alternative explanations is particularly compelling, indicating the need for future research.

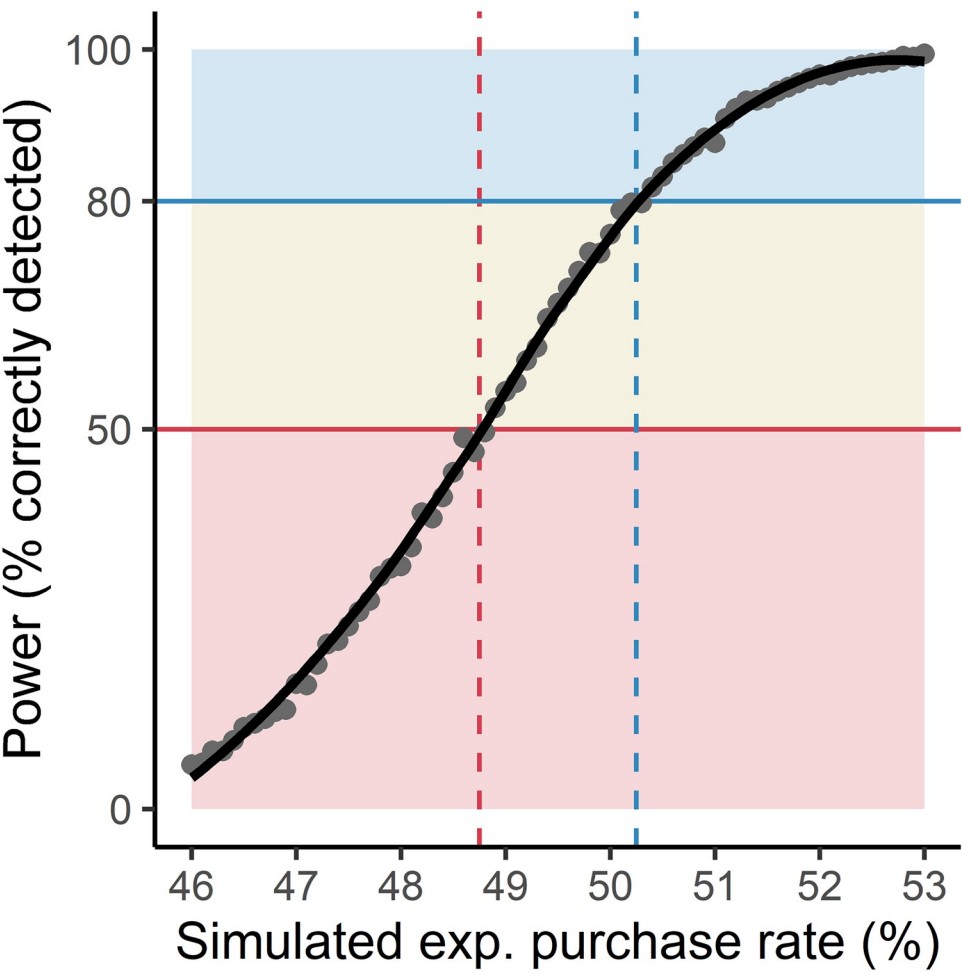

**Fig 4. Results of the posthoc power analysis.** Each dot represents the percentage of correctly detected effects (i.e., statistical power) across 2500 simulated experiments. Each simulated experiment contained 266 fictive participants who had a 45.3% probability of purchasing the control ticket and a variable probability of purchasing the experimental ticket. The solid black line represents a best-fit curve through these points.

## Price-perception effects may have a weaker effect than previously observed in the literature

While previous experiments have relied on stated preferences (i.e., a measure of the participants' willingness to buy differently priced products), our experimental design contained a series of binary purchasing decisions. Such binary purchasing decisions sacrifice the fine-grained continuous measure of stated preference paradigms but benefit from being a conceptually simple and naturalistic measure of purchasing behaviour. These properties are particularly important when conducting online experiments. The sample is unlikely to have previously participated in economic experiments and may inadequately understand the complex incentivization and elicitation methods commonly utilized in previous research [27–29].

Despite their naturalistic and simplistic quality, such binary yes/no purchasing decisions form a double-edged sword. While they allow for a simplistic and naturalistic experimental methodology, it does so at the cost of being a coarse measure of behavior. It is possible that the different price-ending treatments introduced small differences in the perceived attractiveness of the lottery tickets, but that these small differences did not reach a sufficient magnitude to

alter a participants' binary purchasing decision. Our experimental method may be insufficiently sensitive towards small differences in perceived attractiveness, and more fine-grained stated preference may offer a more nuanced set of findings. This conjecture is partially supported by our posthoc power analysis, indicating that we would reliably be able to detect a statistical effect if participants would purchase the experimental tickets in at least 48.75% of decisions. Both the LDE and PFE may have exerted an effect, but at a level below this threshold.

If supported by future research, this interpretation of the data suggests that the strength of both price-perception effects is smaller than previously anticipated–and requires more fine-grained experimental paradigms than the one currently employed in this study. However, this alternative explanation raises questions about the practical relevance of psychological pricing strategies. If the effects of these pricing strategies are indeed highly limited in their ability to influence consumers' purchasing abilities, then their ability to influence consumer behavior may be overshadowed by alternative product-influencing factors. For example, the purchase-enhancing effects of a product's psychological pricing scheme may be easily overwhelmed by alternative environmental factors such as its position in a store, its lighting, or its adjacency to other products. Future research is needed to delineate the scope of decision contexts in which the psychological pricing effect exerts a practical impact on consumer behavior.

Additionally, participants in our experiment evaluated each ticket individually. Under these conditions, a relatively large difference in attractiveness may be needed between the price-ending conditions in order to detect a difference in participants' overall purchasing rates. In line with a more limited effect of price perception, several previous studies had exclusively observed evidence for the LDE when different items were compared and selected in direct competition with one another [30, 31]. Under such a winner-takes-all comparison, relatively tiny differences in the attractiveness between two (or more) alternatives may result in large differences in purchasing rates. For example, during real-world purchasing experiences, such a direct comparison between items may be common–for example, comparing the prices of two adjacent brands of milk in a grocery store. As our experiment does not allow for such decision environments, the question arises whether our individual-ticket based experimental design has a sufficient degree of external validity to allow the generalization of our findings to such comparative purchasing decisions. Future research is required to determine whether either price-perception effects are indeed more likely to occur in comparative decisions than in our current experimental design.

## Price-perception effects may be more pronounced in the direct comparison of items with multiple features

Alternatively, both the LDE and PFE may be particularly pronounced when participants compare items on a range of attributes–of which their price is only one. While our methodology offers a higher degree of naturalism than traditional willingness-to-pay experiments, it falls short of a truly realistic purchasing experience in which customers compare and contrast multiple products based on a range of attributes. While we may naturally capture a sole price-driven purchase, our design may have limited applicability to the multi-attribute nature of decisions encountered in daily shopping experiences.

There is some evidence to support such an influential decision complexity on price perception. The comparison of items with multiple non-price attributes is computationally more complex than a decision between two otherwise homogenous items. This increased complexity places a higher cognitive load on consumers, which may increase consumers' reliance on heuristic decision strategies [32], potentially providing a more fertile ground for both two price

perception effects. At face value, our experimental results do not support such mediation of cognitive load on psychological price-perception. If cognitive load mediates perception effects, then participants with lower numerical ability scores may be expected to have a decreased ability to handle the complexity of the current purchasing task. However, in the current experiment, we did not observe either a LDE or PFE for low-numeracy participants. This lack of differentiation may result from the low cognitive complexity of the lottery task, which may not have posed a sufficient load to spark either of the price-perception effects.

Such an effect of cognitive load on the efficacy of psychological pricing strategies suggests that a complete understanding of price perception requires a detailed account of the consumer's decision context. While our experimental findings are not inconsistent with this conjecture, it also does not provide empirical support for such a mediation effect of cognitive load. Therefore, future research is required to determine whether psychological pricing strategies are more effective in decisions made under a high cognitive load.

## Consumers may have adapted their price heuristics to prevailing marketing conditions

While the above interpretations may explain why our results diverge from previous laboratory studies, they are unable to explain why the LDE and PFE show up in the real world. Field experiments show an increase in a revenue-enhancing effect when companies randomly changed product's pricing schemes to just-below integer levels [8–11]. Furthermore, the long-standing tendency of retailers to utilize both just-below and rounded pricing strategies is presumably based on observed merit. How can these real-world empirical observations be unified with our observed null result in a controlled laboratory environment?

One explanation holds that both seemingly paradoxical observations are true: conceivably, the research outlined during the introduction correctly measured consumers' increased attraction to just-below and rounded pricing schemes. However, these effects may have since stopped existing in the consumer population. Such a disappearance is possible if consumers have learned to compensate for these psychological pricing effects and have adapted their heuristic decision strategies accordingly. After all, retailers have utilized the just-below pricing strategy for multiple decades–allowing entire generations of shoppers to recognize this psychological exploit and circumvent it by making the explicit effort to mentally round prices to their nearest integer. Although circumstantial, the recent increase in the use of rounded prices in the retail sector [5] may hint at such an evolving price perception strategy: consumers may have 'caught on' to the LDE, causing retailers to shift to a different pricing strategy (based on the PFE) instead. Although this interpretation of our results is highly speculative, such an adaptive-heuristic interpretation offers an exciting future venue of research in consumer psychology–suggesting that cognitive mechanisms and their interaction with the environment shape their purchasing behaviour.

## Concluding remarks

In theory, the experimental paradigm developed in this paradigm provided an idealized set of circumstances for both the left digit and perceptual fluency effects: participants made multiple repeated decisions with a highly salient price, and the absence of any non-price attributes prevented a distraction from this price. Our failure to reproduce either effect in these circumstances is surprising and suggests that both effects may be more nuanced or environment-specific than previously discussed. However, these null results provide valuable clues in their own right, providing an avenue for future research to detail when these price-perception effects occur–and when they do not.

## Supporting information

**S1 File. Main results for the excluded participants.** The supporting information contains the results for the 124 participants who purchased one or more of the attention-checks. (DOCX)

## Acknowledgments

The authors are grateful for the helpful comments from the Idealab at the Chair of Finance (Institute for Management Research, Radboud University Nijmegen). We further thank the participants of the 51st Hohenheimer Oberseminar at Heinrich-Heine-Universität Düsseldorf for valuable comments on an earlier version of the paper. Finally, we thank the Faculty of Society and Economics at the Hochschule Rhein-Waal for their ongoing support throughout the research project.

## Author Contributions

**Conceptualization:** Jörn Sickmann, Sascha Füllbrunn, Carina Goldbach, Thomas Pitz.

**Formal analysis:** Achiel Fenneman.

**Investigation:** Achiel Fenneman, Jörn Sickmann, Sascha Füllbrunn, Carina Goldbach, Thomas Pitz.

**Methodology:** Achiel Fenneman, Jörn Sickmann, Sascha Füllbrunn.

**Project administration:** Jörn Sickmann.

**Resources:** Jörn Sickmann, Carina Goldbach, Thomas Pitz.

**Software:** Achiel Fenneman.

**Supervision:** Jörn Sickmann, Sascha Füllbrunn, Thomas Pitz.

**Writing – original draft:** Achiel Fenneman.

**Writing – review & editing:** Achiel Fenneman, Jörn Sickmann, Sascha Füllbrunn, Carina Goldbach, Thomas Pitz.

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
