## [Editor Report · Decision Letter 0]

1 May 2022

PONE-D-22-10700Psychological price perception may exert a weaker effect on purchasing decisions than previously suggested: results from a large online experiment fail to reproduce either a left-digit or perceptual-fluency effectPLOS ONE

Dear Dr. Fenneman,

Thank you for submitting your manuscript to PLOS ONE. I have now read the revised version of your paper (the original version had a different manuscript number). I think the earlier reviewer and editor comments have carefully be considered in the new draft, and I think the paper can be accepted for publication, after some minor edits that I suggest below. I think this is a very well designed and insightful piece of research now. I look forward to the final draft. My comments:- p3 l52: lower valuation of the price: confusing language, what is a valuation of a price. Better rewrite more clearly. - p5 l110: what are compensatory purchasing heuristics. Please explain more carefully. - Footnote 2, excluded participants. Maybe add the basic results for only this group of subjects in the appendix. It may be suggestive of how consumers who do not gather price information well are influenced by the pricing mode. - You sometimes use "rounding treatment" as generic term for the treatments. At the same time you have, indeed, a treatment with rounded numbers. this leat to some confusing for me when reading your paper. Please use a different wording, eg "pricing treatment" or something along these lines. - page 15, l314 course or coarse? - In your discussion of the explanations for your Null effect, I think you can be a bit more explicit that these explanations may not be very compelling. Eg, para ending on p15, l325: if so, what would be the practical relevance of the pricing mode then? Para ending on p15, l334: hat would be the external validity then? Para ending on p16, l351: this would not be consistent with your evidence on numeracy and similar findings for the excluded subjects. - One thing that came to my mind was that the decision mode in shopping may be different: you do not decide buy milk yes or no, but you need buy one milk, but which one do you choose? Then, in direct comparison, pricing mode may become relevant. Maybe this is what you mean with multiple features.  

We look forward to receiving your revised manuscript.

Kind regards,

Stefan T. Trautmann

Academic Editor

PLOS ONE

Journal Requirements:

3. Please remove your figures from within your manuscript file, leaving only the individual TIFF/EPS image files, uploaded separately.  These will be automatically included in the reviewers’ PDF.
---

## [Author Response · Author response to Decision Letter 0]

8 Jun 2022

Dear Dr Trautmann,

Many thanks for the kind words on the manuscript, and the thoughtful suggestions for further improvements. Below is a detailed list of the provided comments and the resulting changes in the manuscript. All suggested modifications to the manuscript have been implemented. Additionally, we made three further modifications to the manuscript, which are discussed at the end of this letter. We hope that this revised and improved manuscript addresses all remaining concerns.

Editor’s comments:

p3 l52: lower valuation of the price: confusing language, what is a valuation of a price. Better rewrite more clearly.

- We have changed “valuation of a price” to the more accurate “valuation of a product”. 

p5 l110: what are compensatory purchasing heuristics. Please explain more carefully.

- The paragraph on page 5 now includes an additional explanation to our previously used phrase “compensatory pricing heuristics”

Footnote 2, excluded participants. Maybe add the basic results for only this group of subjects in the appendix. It may be suggestive of how consumers who do not gather price information well are influenced by the pricing mode.

- An appendix has now been included, which contains the mail results for all excluded participants. As compared to the main sample, these excluded participants have a higher purchasing rate for all lottery tickets. However, as in the main sample, the results from these excluded participants does not provide evidence to support either of the two proposed psychological pricing studies

You sometimes use "rounding treatment" as generic term for the treatments. At the same time you have, indeed, a treatment with rounded numbers. this leat to some confusing for me when reading your paper. Please use a different wording, eg "pricing treatment" or something along these lines.

- In order to reduce this confusing language, “rounding treatment” has now been substituted with “price-ending treatment”

page 15, l314 course or coarse?

- It was indeed coarse, not course. This has been modified in the revised manuscript, of course. 

In your discussion of the explanations for your Null effect, I think you can be a bit more explicit that these explanations may not be very compelling. Eg, para ending on p15, l325: if so, what would be the practical relevance of the pricing mode then? Para ending on p15, l334: hat would be the external validity then? Para ending on p16, l351: this would not be consistent with your evidence on numeracy and similar findings for the excluded subjects.

- We have made the following changes to text to discuss the shortcoming of our proposed alternative explanations. Notably:

o The paragraph starting on line 333 (p15) has been expanded to include a discussion of the limited practical relevance of psychological pricing studies when assuming that our findings reflect a true negative result. 

o The paragraph starting on line 344 (p16) has been expanded to include a discussion of the external validity of our experiment design.

o The paragraph starting on line 367 (p16) has been expanded to include a more detailed discussion regarding the putative effect of cognitive load on price perception effects. Additionally, we include a discussion regarding the inconsistency between this proposed alternative explanation (our design did not induce a sufficient computational load) with the lack of either the LDE or PFE in participants with a low numerical ability score. 

o The paragraph starting on line 379 (p17) has been extended by a brief call for future research into the effects of cognitive load on price-perception effects. 

One thing that came to my mind was that the decision mode in shopping may be different: you do not decide buy milk yes or no, but you need buy one milk, but which one do you choose? Then, in direct comparison, pricing mode may become relevant. Maybe this is what you mean with multiple features. 

- The paragraph starting on line 344 (p16) has been expanded to include a brief discussion of the experiment’s limitation when capturing purchasing decisions for which multiple items are compared side-by-side. 

Additional changes to the manuscript 

In additional to the comments addressed above, we made three additional modifications to the manuscript. In particular, these changes include: 

- A detailed inspection of the data revealed a previously un-caught error in the first panel of figure 2. In the original manuscript, the purchasing percentage of all three lottery tickets was depicted incorrectly. This error only applied to the graph itself, not to the underlying data – hence, none of the formal tests were impacted. In the revised manuscript we modified this image. For conceptual completeness, we re-ran the power-simulation using the “new” baseline purchasing rates (45.3% instead of the erroneous 45.9%). This did not result in a meaningful difference between the original manuscript and the current version. The analysis script has been carefully checked to exclude any potential other errors. 

- A brief paragraph has been added to the Methods section with a description of the ethical approval process, anonymization of participant’s data and the informed consent procedure. 

- The Acknowledgements have been updated with a line thanking the Hochschule Rhein-Waal for their financial support.

---

## [Editor Report · Decision Letter 1]

20 Jun 2022

Psychological price perception may exert a weaker effect on purchasing decisions than previously suggested: results from a large online experiment fail to reproduce either a left-digit or perceptual-fluency effect

PONE-D-22-10700R1

Dear Dr. Fenneman,

We’re pleased to inform you that your manuscript has been judged scientifically suitable for publication and will be formally accepted for publication once it meets all outstanding technical requirements.

Kind regards,

Stefan T. Trautmann

Academic Editor

PLOS ONE
---

## [Editor Report · Acceptance letter]

10 Aug 2022

PONE-D-22-10700R1 

Psychological price perception may exert a weaker effect on purchasing decisions than previously suggested: results from a large online experiment fail to reproduce either a left-digit or perceptual-fluency effect 

Dear Dr. Fenneman:

I'm pleased to inform you that your manuscript has been deemed suitable for publication in PLOS ONE. Congratulations! Your manuscript is now with our production department. 

Kind regards, 

on behalf of

Professor Stefan T. Trautmann 

Academic Editor

PLOS ONE